# Investigating the Index of Sustainable Development and Reduction in Greenhouse Gases of Renewable Energies

**Vadim V. Ponkratov [1], Alexey S. Kuznetsov [2] , Iskandar Muda [3,\*] , Miftahul Jannah Nasution [4], Mohammed Al-Bahrani [5] and Hikmet Ş. Aybar [6,7,\*]**

1. Department of Public Finance, Financial University under the Government of the Russian Federation, 125167 Moscow, Russia
2. Laser and Optoelectronic Systems Department, Bauman Moscow State Technical University, 105005 Moscow, Russia
3. Department of Doctoral Program, Faculty Economic and Business, Universitas Sumatera Utara, Medan 20222, Indonesia
4. Badan Penelitian dan Pengembangan Daerah Provinsi Lampung, Kota Bandar Lampung 35212, Indonesia
5. Chemical Engineering and Petroleum Industries Department, Al-Mustaqbal University College, Babylon 51001, Iraq
6. Department of Mechanical Engineering, Eastern Mediterranean University, TRNC, Via Mersin 10, Famagusta 99628, Turkey
7. Department of Medical Research, China Medical University Hospital, China Medical University, Taichung 404, Taiwan
* Correspondence: iskandar1@usu.ac.id (I.M.); hikmet.aybar@emu.edu.tr (H.Ş.A.)

**Abstract:** Considering the limited resources of fossil energy and the problems caused by the emission of greenhouse gases, it is necessary to pay more attention to renewable energies, because in this way, the goals of sustainable development can be achieved. The importance of renewable energies in sustainable development, reducing greenhouse gases and increasing energy security on the one hand, and the need for financial resources and large investments for renewable energy projects on the other hand, doubles the role and importance of financial development in the development of renewable energies. Considering the importance of this issue, the present study examines the impact of the development of modern facilities and renewable energy technology. In this study, dynamic interactions in the Sustainable-Energy-Energy Development Pattern of carbon dioxide are investigated using the Bayesian Vector Auto Regression (BVAR) method. One of the most important indicators for evaluating sustainable development is the modified pure arrangement (GS). For this purpose, this index was used as a sustainable development index. The results indicate that the effect of positive impulse on renewable and renewable energy consumption on sustainable development in Uganda is positive. In addition, the positive shock of renewable and renewable energy consumption increases the emissions of carbon dioxide pollutants to a different extent. In addition, the effect of the growth of sustainable development index on renewable energy consumption and renewal energy consumption is ($CO_2$) negative. The research results show that based on the RMSE criterion, the former SSVS-Full function was used to investigate the impact of renewable energy consumption on sustainable development and the independent Normal-Wish art function was used. Therefore, in this research, the dynamic relationships between sustainable development, energy consumption (separately from renewable and non-renewable energy) and $CO_2$ emissions are investigated.

**Keywords:** sustainable development; renewable energy; greenhouse gases

## 1. Introduction

Energy is a key factor in economic development, but sustainable growth cannot be achieved without environmental protection and economic growth [1–3]. Any structural reforms that reduce a country's energy consumption may reduce economic growth. Saving energy consumption and developing and applying alternative technologies, especially

renewable energies, can help control and reduce the consumption of fossil energy carriers, reducing pollution and achieving sustainable development [4,5]. Sustainable development is an evolutionary process because it combines economic growth, environmental protection, and social justice across generations [6–8]. Sustainable development aims to balance economic, social, and environmental dimensions, contrary to popular belief [2].

The replacement of primary energy sources that emit less pollution in the environment has become an issue in economic growth [9,10]. Most countries in the world are taking steps to encourage people and economic institutions to use renewable energy sources. In Uganda, the Ministry of Energy and the Ministry of Petroleum are required by law to support the use of renewable energy sources. Uganda's potential for using renewable resources is high, but it has not been fully exploited until now [11,12].

In the early 1980s, statesmen realized that knowing the cause-and-effect relationships between energy consumption, environmental quality, and economic growth and development is the first step in optimally explaining energy policies. Four hypotheses explain the relationship between energy consumption and economic growth: growth, resource conservation, feedback, and neutrality [13]. Preserving natural resources implies a one-way causal link between economic growth and energy consumption. This hypothesis says resource-saving policies will not affect economic growth. The growth hypothesis says energy consumption causes economic growth. Energy conservation may slow economic growth. The feedback hypothesis implies a two-way causal relationship between energy consumption and economic growth [14]. Therefore, policies to expand energy consumption have a positive effect on economic growth and vice versa [15]. The neutrality hypothesis states that the energy consumption shock has no effect on economic growth. Energy-saving or energy-development policies do not affect economic growth [15]. It is necessary to achieve the goal of economic stability, stabilization, or reduction in greenhouse gas emissions [16]. This requires a transition from economic activities based on polluting energy sources to sustainable economic activities based on technologies and consumption with less environmental impact [3]. Humans have always valued nature, so protecting biodiversity is essential. Reducing greenhouse gas emissions is necessary to accomplish this. Since carbon emissions play a large role in the production of greenhouse gases, factors affecting their emission must be investigated. Greenhouse gases cause global warming. This is why renewable energy use must grow [17]. Climate change has been a major challenge since the beginning of this century. For this reason, many studies have discussed the need to reduce $CO_2$ emissions that have disrupted human existence [14].

Energy consumption and economic growth are major causes of greenhouse gas emissions, according to empirical research. Ecological economists say energy is the only and most important growth factor in the biophysical model [17]. Therefore, according to them, labor force and capital need energy. Some economists say energy affects economic growth indirectly through labor force and capital, but not directly [18]. These economists believe that energy is an intermediary input and the basic factors of production are only labor, capital, and land. However, excessive consumption of energy, especially fossil fuels, to achieve the goal of economic growth has caused an increase in environmental pollution [19]. The environmental consequences of global warming, climate change, and greenhouse gas emissions have increased concerns about non-renewable energy consumption and public interest in renewable energy has grown because it reduces greenhouse gas emissions, reducing carbon dioxide emissions and protecting the environment [20]. Renewable energy reduces greenhouse gas emissions, high prices, volatile energy, and foreign energy dependence [21]. Technical innovation on the environment has been ongoing in recent years. The endogenous growth literature shows that technical innovation can have a positive effect on the environment in the long term [16]. According to these economists, technical innovation directly and indirectly increases investment by reducing the cost of information and exchanges, increasing the productivity of production factors, increasing savings and improving resource allocation; it is eco-friendly [22]. In contrast to this view, some economists believe that technical innovation can harm the environment [23]. If technical

innovation improves resource allocation and saving efficiency, the savings rate may drop, leading to a credit crisis and harming the environment by reducing investment [24,25]. Environmental pollution is a challenging topic, so many researchers have studied it. Different methods, approaches, and samples have yielded contradictory results and research is ongoing [26,27].

This study tries to overcome previous studies' flaws. This research's most innovative aspects are: First, an efficient index is used to express sustainable development. Second, renewable and non-renewable energy are separated to investigate the relationship between sustainable development, air pollution, and energy consumption using different types of energy. Third, the Bayesian Vector Auto Regression (BVAR) method is used to examine the dynamic interrelationships between energy consumption and sustainable development. In this research, efficiency index was used as a sustainable development index. The impact of positive incentives on renewable energy consumption on sustainable development in Uganda is investigated. Further, the positive shock of renewable energy consumption increases the emission of carbon dioxide pollutants to a different extent. In addition, the impact of sustainable development index growth on renewable energy consumption and renewable energy consumption has been evaluated.

## 2. Method

Basically, the energy supply of greenhouses, or in other words, the provision of electrical, cooling, and heating loads required by greenhouses for the four main components of lighting, internal temperature, $CO_2$ emission, and relative humidity [28]. If the greenhouse is connected to the main grid using electricity and gas, it can supply the electricity needed for its electrical, cooling, and heating equipment [29]. However, it can be checked if renewable energy units are used together with thermal generators, to what extent the energy supply of greenhouse gases during the day by these energy units and thermal generators instead of buying electricity and producing electricity is affordable [22]. Gas from the main grid should be cost effective; in other words, the energy supply of buildings, for example, greenhouses, which need more energy, with the benefit of renewable sources as the main sources and the main power of the grid as a backup source is considered in this matter and has been by creating such a mechanism, in addition to supplying the energy needed by such structures, energy exchange between renewable sources and the main grid is established in the form of buying and selling production power [30]. The described system for building energy supply can be seen in Figure 1, where for the sake of easier description, considering the direct connection of renewable energies with greenhouse gas emissions, the energy supply of a greenhouse is set as a criterion [31].

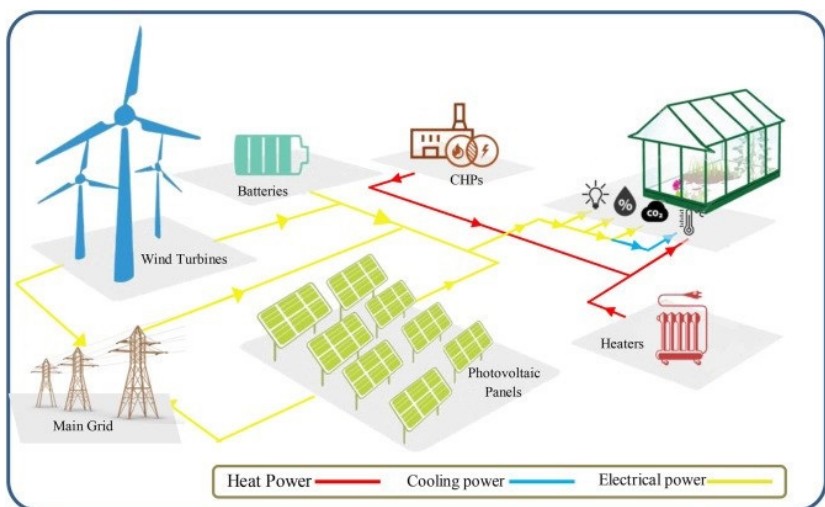

**Figure 1.** Energy exchange between renewable energies, the main energy grid and buildings. Yellow axis: energy consumption, blue axis: energy recycling, red axis: energy consumption.

As can be seen in Figure 1, if the energy requirements of the buildings are supplied by renewable units and storage heaters, in addition to the income from savings due to not buying power from the main grid, in some cases, surplus can also be obtained. He sold the generated electric power to the main grid and earned other income. However, the realization of these incomes depends on the number of energy units and storage devices. Therefore, the purpose of energy planning of structures and buildings using renewable resources is to determine the optimal number of energy units and storage devices according to cost and income parameters and limitations. By determining the optimal number of energy units and storage devices and determining the production power of renewable resources in each hour of planning, four different modes can occur according to the amount of energy supply required: selling power to the main grid, buying power from the main grid, power storage in storage devices, and load supply by storage batteries [32].

The model presented in this research, inspired by the studies conducted, is defined as the following equation:

$$
\begin{bmatrix} GS_t \\ EC_t \\ CO_{2_t} \end{bmatrix}' = z_t' C + \sum_{j=1}^{\rho} \left( \begin{bmatrix} GS_{t-1} \\ EC_{t-1} \\ CO_{2_{t-1}} \end{bmatrix}' Aj \right) + \begin{bmatrix} \varepsilon_t^{GS} \\ \varepsilon_t^{EC} \\ \varepsilon_t^{CO_2} \end{bmatrix}', \tag{1}
$$

where $GS_t$ sustainable development index per capita in terms of dollars, $EC_t$ energy consumption index (renewable and non-renewable) per capita is equivalent to kilograms of oil and $CO_{2_t}$ $CO_2$ emission index per capita in metric tons. All variables are entered in logarithmic form in the model and Vector $z_t'$ the vector of fixed components and $\varepsilon_t^{EC}$ are the error components of the model. Sustainable Development Index is based on the Green Data Book

$$
GS_t = GNS_t - Ncd_t - Fcd_t - Env_t + Edu_t \tag{2}
$$

In Equation (2) $GS_t$ sustainable development index, $GNS_t$ gross national savings, $Ncd_t$ is the cost of reducing natural resources, $Fcd_t$ is the cost of reducing physical capital, $Env_t$ is the cost of environmental pollution, and $Edu_t$ is the cost of education. All these variables are per capita and in dollars.

In this research, the BVAR method was used to investigate the effect of model variables on each other. One of the main advantages of this method is that it solves the basic problem of unrestricted vector auto-regression models, which is the multiplicity of parameters. This method is especially useful in cases where there are data limitations (such as the economy of Uganda) in shrinking vector auto-regression models and removing the uncertainty of these models. Further, the impact of energy consumption on sustainable development is estimated by separating renewable and non-renewable energy and the results together [33]. Therefore, the estimates in this research include three models as follows:

1. Sustainable development and renewable energy consumption (GS and EC-RE).
2. Sustainable development and non-renewable energy consumption (GS and EC-NRE).

All Bayesian models have three basic components: prior density function, exponential function, and posterior density function. Different results can be obtained by using different types of previous function [2]. Therefore, choosing the appropriate prior function in Bayesian models is very important. Several prior functions have been used in Bayesian vector auto-regression models, which are introduced below.

### 2.1. Former Function of Minnesota

In this type of previous function, it is assumed that the matrix is a diagonal matrix. In this case, each of the VAR equations can be estimated separately using the OLS method. The only task of the researcher in this method is to determine the initial function for vector of model coefficients, which is considered as Equation (3).

$$
\alpha \sim N(\underline{\alpha}_{min} . \underline{v}_{min}) \tag{3}
$$

### 2.2. Prior Function of Natural Twins

These types of functions are such that they cause the distribution of the prior, right and posterior function to be of the same family. The prior function of natural twins is as follows:

$$\alpha \Big| \sum \sim N\Big(\underline{\alpha}, \sum \otimes \underline{V}\Big) \tag{4}$$

$$\sum{}^{-4} \sim W\Big(\underline{S}^{-1}, \underline{v}\Big) \tag{5}$$

so that $\underline{\alpha}$, $\underline{V}$, $\underline{v}$, and $\underline{S}$ are hyperparameters that the researcher chooses.

### 2.3. Independent Wishart Prior Function

Wishart's independent prior function is a general prior function for the model. This function can be written as follows:

$$\rho\Big(\beta, \Sigma^{-1}\Big) = \rho(\beta)\rho\Big(\Sigma^{-1}\Big) \tag{6}$$

$$\beta \ \sim \ N\ \Big(\underline{\beta}, \underline{V\beta}\Big) \tag{7}$$

$$\sum{}^{-4} \sim W\Big(\underline{S}^{-1}, \underline{v}\Big) \tag{8}$$

### 2.4. SSVS-Wishart Prior Function

The SSVS prior function can be defined as a hierarchical prior function in the form of Equation (9).

$$\alpha | \gamma \sim N(0, \ DRD) \tag{9}$$

In such a way, matrix $R$ is the previous correlation matrix, which for simplicity is usually considered equal to matrix $I$ and it expresses the basic belief that there is no correlation between the coefficients of the model, and $D$ is a diagonal matrix where $(j,j)$ the elements are equal to $d_j$ is Equation (10).

$$d_j = \begin{cases} k_{0j} & if & y_j = 0 \\ k_{1j} & if & y_j = 1 \end{cases} \tag{10}$$

### 2.5. SSVS-Full Previous Function

In this method, in addition to the model coefficients, the variance in the error components also has the SSVS prior distribution. Considering—as a high-triangular matrix—conversely, the variance–covariance matrix of the error components can be written as follows:

$$\sum{}^{-1} = \Psi' \Psi \tag{11}$$

According to the prior function SSVS the square of each element of the principal diameter ($\Psi$) has a standard gamma prior distribution and the elements above the principal diameter of this matrix have a hierarchical distribution similar to Prior distribution $\alpha$ will have.

## 3. Results and Discussion

In this research, at first, the stationarity of the variables was investigated using unit root tests and the results are presented in the table below.

The results of Table 1 show the instability of the model variables at the level and the instability of their first-order difference. If there is a collinear relationship between the variables, the VECM model should be used. However, since the VECM model can be written in an equivalent form by changing the parameters, if there is a co-occurrence relationship between the variables in a model, a VAR model of unstable variables can be estimated and valid results can be obtained (Figure 2).

**Table 1.** Results of stability tests of variables.

| Possibility PP | Possibility ADF | First Order Difference | Possibility PP | Possibility ADF | Level |
|---|---|---|---|---|---|
| −4.32 (0.151) | −6.24 (0.1) | D (GS) | −0.40 (0.63) | −0.41 (0.58) | GS |
| −3.25 (0.151) | −3.25 (0.151) | D (EC-re) | −0.25 (1.34) | −0.23 (1.23) | EC-re |
| −9.35 (0.151) | −8.15 (0.01) | D (EC-nre) | −2.15 (0.63) | −2.12 (0.63) | EC-nre |
| −4.98 (0.151) | −2.39 (0.151) | D ($CO_2$) | −4.10 (1.41) | −4.15 (1.65) | $CO_2$ |

It has been used to check the existence of collinearity between variables and to determine the number of convergence vectors. The results of the pooled test are listed in Table 2.

**Table 2.** Results of cointegration test.

| Possibility | Critical Values (0.5) | The Value of the Statistic λ Trace | The Number of Collocation Vectors |
|---|---|---|---|
| | | | GS, EC-re, $CO_2$ |
| 0.95 | 18.52 | 20.45 | r = 0 |
| 0.021 | 15.32 | 12.36 | r ≤ 1 |
| 0.65 | 10.89 | 8.30 | r ≤ 2 |

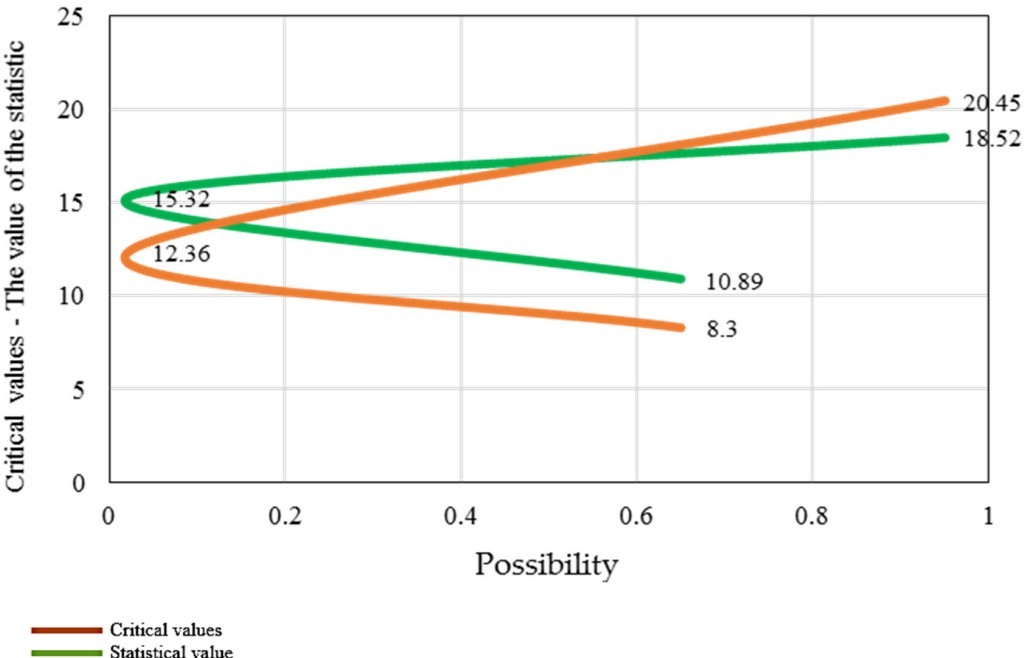

**Figure 2.** The indices of homogeneity test in critical values and statistical value.

Table 3 shows model variables co-occurring. Non-renewable energy has also confirmed the co-accumulation relationship between model variables. Models can be estimated for different intervals to determine the optimal interval length based on information criteria. Sustainable development is the optimal $CO_2$ emission interval length for renewable and non-renewable energy models.

Out-of-sample forecasting is one way to compare Bayesian vector auto-regression model efficiencies and choose the best prior function. RMSE checks model predictions. Different prior functions and prediction horizons have different RMSE values. Using the SSVS Full prior function, the BVAR model provides more accurate predictions. Due to lower prediction error, the first model will use the Bayesian method with the SSVS Full prior function (renewable energy). Based on repeated calculations, the second model will use the Wishart prior function (non-renewable energy). Independently, it is truer.

**Table 3.** RMSE index prediction of model (GS, EC-re, $CO_2$) by different prior function.

| Relative Index | Average 1 to 4 Courses | h = 4 | h = 3 | h = 2 | h = 1 | The Previous Function Type |
|---|---|---|---|---|---|---|
| 0.36 | 0.198 | 0.035 | 0.158 | 0.265 | 0.021 | Scattered |
| 0.52 | 0.264 | 0.232 | 0.324 | 0.214 | 0.235 | Minnesota |
| 0.95 | 0.189 | 0.148 | 0.148 | 0.219 | 0.015 | Natural twins |
| 0.69 | 0.236 | 0.325 | 0.154 | 0.236 | 0.365 | Normal-Independent Wishart |
| 0.84 | 0.301 | 0.068 | 0.328 | 0.258 | 0.314 | SSVS-Wishart |
| 0.74 | 0.201 | 0.214 | 0.158 | 0.321 | 0.032 | SSVS-Full |

In vector auto-regression models, instantaneous reaction functions are used to investigate the effect of creating a shock in a specific variable and the reaction of other variables in the model. Investigating instantaneous reaction functions is actually the same as studying the timing of the effect of impulses. In these functions, the effect of creating a shock equal to one standard deviation in one variable on other variables in the model is investigated. To calculate instantaneous reaction functions in Bayesian econometrics, posterior simulation methods are used. In order to check the significance of response functions, confidence bands have been calculated using sampling simulation. Significance in instantaneous response functions means that the response of the relevant variables is not statistically zero and this problem occurs when the confidence bands are placed on one side (horizontal axis).

As can be seen in Figure 3, the impact of energy consumption shock on sustainable development is positive, regardless of the type of energy; in other words, increasing the consumption of renewable and non-renewable energy will lead to the growth and improvement in sustainable development in Uganda. Creating a boost in renewable energy consumption increases sustainable development from the first period and after reaching the maximum in the 10th period, it slowly slows down. Further, the effect of increasing non-renewable energy consumption on sustainable development appeared after 8 periods and reached its peak in the 25th period and then gradually subsided. The result obtained is consistent with the results of studies that have investigated the positive impact of renewable and non-renewable energy shocks on economic growth in different countries. Because economic growth followed by an increase in gross national savings is one of the factors influencing sustainable development, the factors that cause rapid economic growth can put the country on the path of promoting sustainable development in the short term. However, it should be noted that in the long term, one-sided attention to economic growth and excessive consumption of fossil energy will have destructive environmental consequences due to the lower cost of installation and operation than renewable energy, which will make sustainable development difficult.

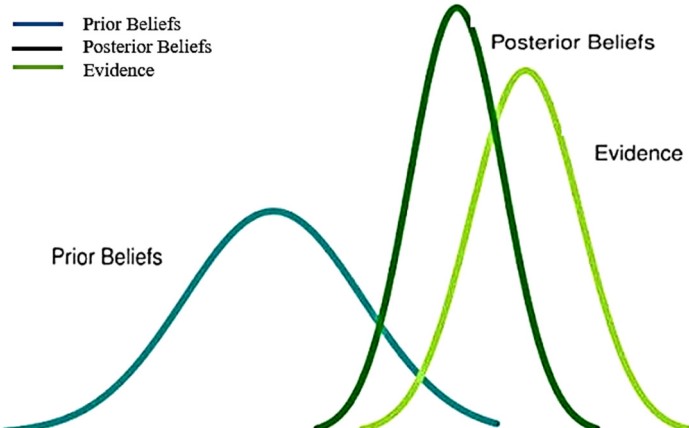

**Figure 3.** Normal distribution of indicators related to previous functions.

According to Figure 4, above, the positive shock effect of sustainable development is positive on renewable energy consumption and negative on non-renewable energy consumption. Further, the reaction of renewable and non-renewable energy consumption to the shock of sustainable development is not permanent and gradually fades. The results of studies on energy consumption and economic growth indicate that the positive shock of economic growth has had a positive effect on the consumption of renewable and non-renewable energy. The results obtained from the relationship between sustainable development and energy consumption are different. Economic growth does not pay attention to the harmful environmental effects of the consumption of energy resources and the increase in economic growth leads to the growth of physical capital and, as a result, more investment in the exploitation of energy resources, especially fossil energies. However, the positive shock of sustainable development and achieving higher stages of sustainable development in the country will increase the focus on the quality of the environment and social welfare, which are other components in sustainable development and planning to encourage investment in renewable energy and reduce the consumption of fossil energy.

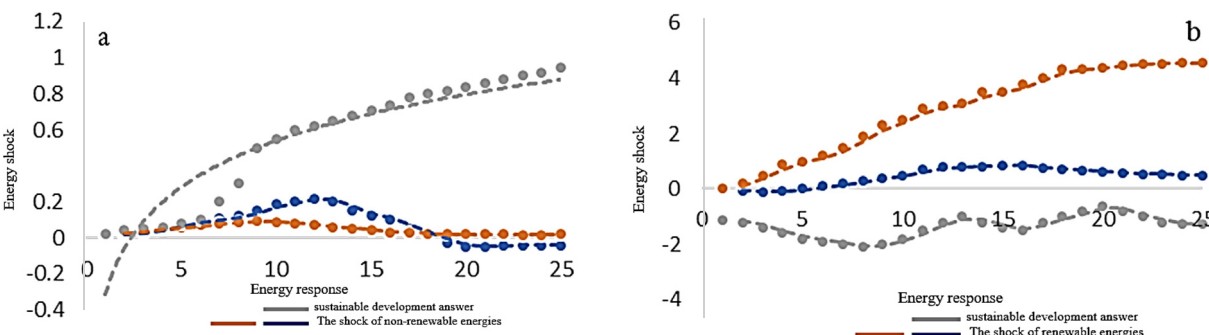

**Figure 4.** The response of sustainable development to energy consumption shocks during the period. (**a**) Sustainable development response to renewable energy shock, (**b**) sustainable development response to non-renewable energy shock.

Figures 5 and 6 show the distributional effect of shocks to renewable and non-renewable energy variables on the consumption of $CO_2$ pollution. Based on these graphs, it can be concluded that the increase in renewable energy consumption initially causes an increase in the emission of carbon dioxide pollutant, but after 5 periods and reaching the maximum point, air pollution decreases and, after 21 periods, it subsides; this is despite the fact that the increase in non-renewable energy consumption has aggravated air pollution and this effect disappears after 24 periods.

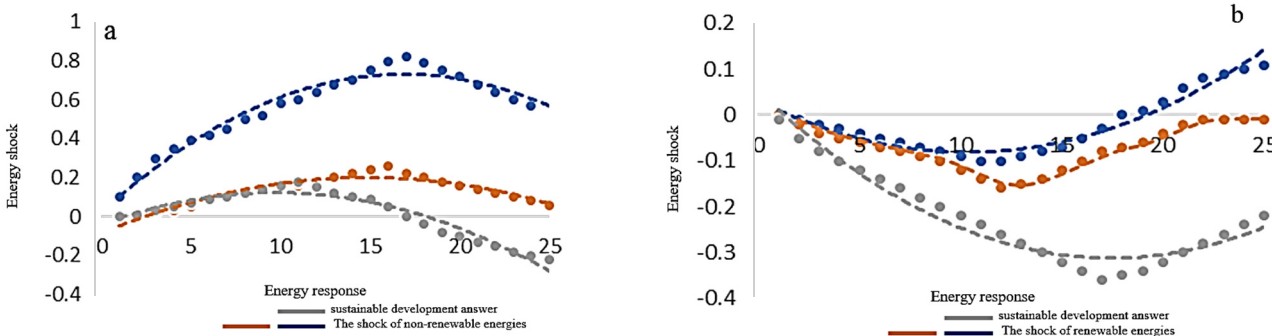

**Figure 5.** The response of energy consumption to the shock on sustainable development during the period. (**a**) The response of renewable energy to the shock of sustainable development, (**b**) the response of non-renewable energy to the shock of sustainable development.

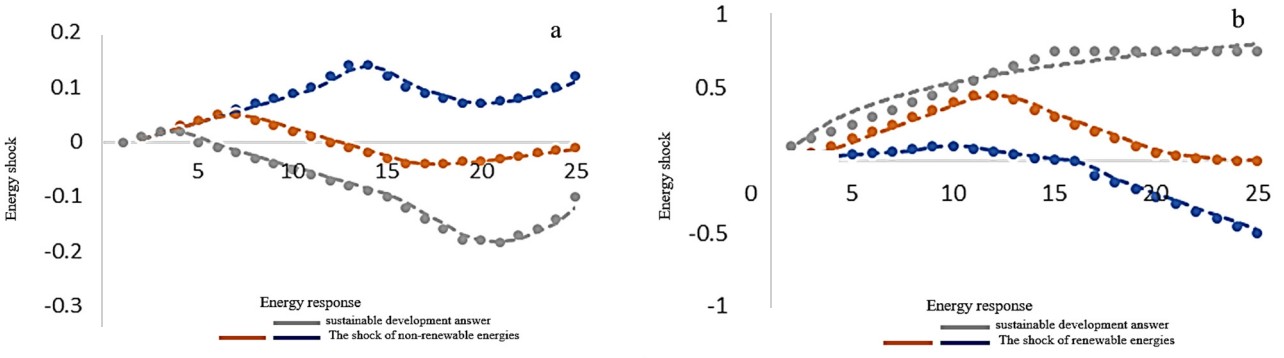

**Figure 6.** Response of $CO_2$ emission to shock of renewable and non-renewable energy over the period. (**a**) Response of $CO_2$ emission to renewable energy shock, (**b**) response of $CO_2$ emission to non-renewable energy shock.

The use and exploitation of renewable energies brings benefits, such as being infinite, clean and renewable, cost effective, reducing air pollution, and also reducing the emission of greenhouse gases. In this research, renewable energies were examined and studied and the generalities of the research are as follows: 1. Solar energy is the heat and light emitted by the sun. 2. Wind energy is energy from moving air. When the sun's radiation reaches the uneven surfaces of the earth unevenly, it causes changes in temperature and pressure and, as a result of these changes, wind is created. 3. Hydrogen (H2) is obtained from some raw materials, such as natural gas, coal, and water. Hydrogen is a colorless, odorless, and non-toxic gas, hydrogen flames are invisible and smokeless. 4. Waves are created by the transfer of energy from wind to water. Wave energy has advantages, such as renewable energy, abundance of energy, naturalness of this energy, reduction in dependence on fossil energy, non-pollution of the environment by it, stability and compatibility with the environment, high concentration capability, and ability to install equipment, dispersion, and abundance of wave energy. 5. Tidal energy is created by the attraction of the moon and the sun during the rotation of the earth. 6. Hydropower is hydraulic energy that can store kinetic energy into potential energy with the aim of producing and supplying a percentage of the electricity consumed in the country or regulating the grid during peak consumption hours by building dams and creating hydropower plants to meet this need slowly. 7. Geothermal energy is the energy available deep in the earth, which originates from the solar energy that has been stored inside the earth for thousands of years, as well as the decay of radioactive uranium, thorium, and potassium isotopes over many years deep in the earth. It is mainly concentrated in earthquake-prone and young volcanic areas and tectonic plates of the earth. 8. Biomass is considered one of the important sources of renewable energy and includes forests, plant parts, leaves, living organisms of the oceans,

animal waste, urban, and food waste, etc. Biomass has the ability to produce electricity, heat, liquid fuels, gaseous fuels, and various useful chemical applications. The most important energy sources are renewable and renewable energy sources are shown in Figure 7.

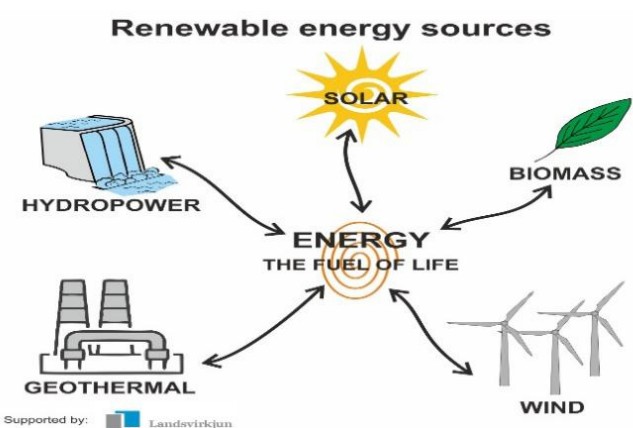

**Figure 7.** The most important energy sources renewable energy sources.

Due to the limitation of fossil energy resources and the problems caused by the emission of greenhouse gases, it is necessary to pay more attention to renewable energy, because, in this way, the goals of sustainable development can be achieved. The importance of renewable energy in sustainable development, reducing greenhouse gases and increasing energy security on the one hand, and the need for financial resources and large investments for renewable energy projects on the other hand, doubles the role and importance of financial development. Renewable energy development considering the importance of this issue, the present study examines the impact of the development of modern facilities and renewable energy technology.

### 4. Conclusions

Due to climate change and global warming, sustainable development is crucial. Based on the RMSE criterion, the former SSVS-Full function and the independent Normal-Wish art former function are used to investigate the effect of renewable energy consumption on sustainable development. This research investigates the dynamic relationships between sustainable development, energy consumption (renewable and nonrenewable), and $CO_2$ emissions in Uganda from 1980–2013. Ani is used to study the effects of shocks on model variables. BVAR is better than other functions for investigating non-renewable energy consumption and sustainable development. Instantaneous response function results are:

According to the results, renewable and non-renewable energy have a positive impact on Uganda's sustainable development. Sustainable development's response to non-renewable energy is meaningless in the period under review, while its response to renewable energy is significant and positive from period 1 to 20 and meaningless thereafter. Sustainable development is increasingly sensitive to renewable energy use. Renewable and nonrenewable energy consumption react differently to sustainable development's positive shock. Sustainable development in Uganda will increase renewable energy consumption and decrease non-renewable energy consumption. To achieve sustainable development, especially in developing countries, such as Uganda, it is necessary to use all types of energy (renewable and fossil). It achieved sustainable development despite environmental damage. The results can be summarized as follows: although renewable and non-renewable energy consumption increase sustainable development in the country, promoting sustainable development increases the tendency to use non-renewable energy. Renewable energy sources (such as wind and sun) are not always available and large-scale storage is impossible, causing low capacity and reduction. These issues show the need to support non-renewable production capacities for energy supply stability. Clean energy is a better choice to increase

stability, but if the government focuses on rapid economic growth and increasing gross national savings, non-renewable energy is chosen due to its lower cost.

According to research results and the need to use non-renewable resources in achieving sustainable development in a country, replacing non-renewable energy with non-renewable resources in the country should be done slowly and with a gentle slope. A sudden decrease in fossil resource use does not help the country's development. Due to the high costs of installing and operating renewable technologies and their insignificant contribution to the country's energy portfolio, it reduces economic growth and sustainable development. Renewable energy will improve Uganda's sustainable development, but it will also cause air pollution. However, this pollution is less than that from non-renewable energy and will disappear faster. Renewable energies cause less pollution than non-renewable energies, but they do not produce zero pollution. According to the results, renewable energy sources are preferable to non-renewable sources for reducing air pollution.

The sustainable development index is calculated using gross national savings. In the short term and early stages, it is important to grow this component for development because this goal requires all types of energy (renewable and non-renewable). After increasing GNP and moving toward sustainable development, it is important to reduce natural resources, especially non-renewable energy and smog. Sustainable development index growth boosts renewable energy investment and consumption. GDP measures financial changes but not future challenges, based on the topics mentioned. Macro decisions should use the adjusted net savings index. GS as a macroeconomic indicator can lead policymakers to protect the environment, prioritize natural resources, and improve society's well-being. "Green GDP" can be used to plan economic growth. Economic progress and an increase in gross national savings can lead to policies that hinder sustainable development. Long-term sustainable development requires policymakers to diversify energy supplies, especially renewables. Create a fund to grant loans for renewable power plant construction and support private investors to promote renewable energy. Fossil resources can replace them.

- Policy implication/suggestions

The importance of renewable energy in sustainable development, reducing greenhouse gases and increasing energy security on the one hand, and the need for financial resources and large investments for renewable energy projects on the other hand, doubles the role and importance of financial development. For renewable energy development, considering the importance of this issue, the present study examines the impact of the development of modern facilities and renewable energy technology.

**Author Contributions:** Conceptualization, V.V.P., I.M. and H.Ş.A.; Data curation, M.J.N.; Formal analysis, V.V.P. and I.M.; Investigation, A.S.K. and M.A.-B.; Methodology, A.S.K.; Resources, I.M. and H.Ş.A.; Validation, M.J.N.; Visualization, M.A.-B.; Writing—original draft, M.A.-B.; Writing—review & editing, H.Ş.A. All authors have read and agreed to the published version of the manuscript.

**Funding:** This research received no external funding.

**Data Availability Statement:** Not applicable.

**Conflicts of Interest:** The authors declare no conflict of interest.

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
