# Peer review of "Investigating the Index of Sustainable Development and Reduction in Greenhouse Gases of Renewable Energies"

_sustainability, doi:10.3390/su142214829_

Round 1

Reviewer 1 Report

Authors investigate the impacts of the integration of renewable energy technologies by focusing on corresponding indices. The topic is interesting and worth to be investigated. This reviewer has the following comments:

1.       The title of this paper could be more concise, reflecting the background, methodology, and target.

2.       This manuscript needs substantial proof-reading. There are a number of grammar and spelling mistakes, e.g.  ‘slow Considering’.

3.       The research gaps in existing works need to be identified and contributions of this work needs to be clearly defined. From this reviewer’s point of view, this paper only investigates indices affected by renewables. What is novelty of this work and new findings?

4.       In Fig.1 what different colour of flows indicate needs to be defined.

5.       Legends should be added to all the figures.

Author Response

Thank you very much for your message regarding this article and permission to review our article. We also appreciate the reviewers' comments, which were very helpful in revising the paper, and we have done our best to address the issues mentioned. In this regard, major corrections were made in our paper, and all corrected items are highlighted in gray in the revised manuscript. Below are the answers to the relevant comments.

Reviewer 2 Report

+ improve the quality of the keywords and follow the policy of the journal as well.

+improve the introduction section with more facts and figures regarding the underline topic with suitable citations in support.

+ well define the abbreviations before their first use. check the whole manuscript thoroughly for this purpose.

+conclused the results section with the summarization of the empirical outcomes.

+add a separate section of discussion for the common/non-technical reader of this field

+Strict with the outcomes of the analysis in the conclusion section. No need to be generalized and the focus should be on the analysis outcomes.

+ add a separate section of "Policy implication/suggestions at the end of the conclusion.

Author Response

(The authors gave the same response as above.)

Reviewer 3 Report

The manuscript needs following revisions to become acceptable for publication:

1. It is better to add some important findings, preferably quantitative, in abstract section. 

2. Importance and novelties of the work must be clearly reflected and denoted. 

3. Introduction section is too long. It is better to shorten it and just reflect the most relevant ones. 

4. If the figures are reprinted, permission is required. 

5. Add reference for the equations that are not derived by the authors. 

6. Adding nomenclature is suggested. 

7. Conclusion is too long. 

8. Following references are suggested to be added in order to improve litertaure review:

"Techno-Economic and Environmental Analysis of Floating Photovoltaic Power Plants: A Case Study of Iran" 10.22044/RERA.2022.11726.1104

"Applying wind energy as a clean source for reverse osmosis desalination: A comprehensive review" https://doi.org/10.1016/j.aej.2022.06.056

"Applications of multi-Criteria Decision-Making (MCDM) Methods in Renewable Energy Development: A Review" https://dx.doi.org/10.22044/rera.2020.8541.1006

Author Response

(The authors gave the same response as above.)
